# The Chloroplast Envelope of Angiosperms Contains a Peptidoglycan Layer

**DOI:** 10.3390/cells12040563

**Published:** 2023-02-09

**Authors:** Xuan Tran, Erva Keskin, Paul Winkler, Marvin Braun, Üner Kolukisaoglu

**Affiliations:** Center for Plant Molecular Biology (ZMBP), University of Tübingen, 72076 Tübingen, Germany

**Keywords:** chloroplasts, plant peptidoglycan, endosymbiotic theory, click chemistry

## Abstract

Plastids in plants are assumed to have evolved from cyanobacteria as they have maintained several bacterial features. Recently, peptidoglycans, as bacterial cell wall components, have been shown to exist in the envelopes of moss chloroplasts. Phylogenomic comparisons of bacterial and plant genomes have raised the question of whether such structures are also part of chloroplasts in angiosperms. To address this question, we visualized canonical amino acids of peptidoglycan around chloroplasts of *Arabidopsis* and *Nicotiana* via click chemistry and fluorescence microscopy. Additional detection by different peptidoglycan-binding proteins from bacteria and animals supported this observation. Further *Arabidopsis* experiments with D-cycloserine and AtMurE knock-out lines, both affecting putative peptidoglycan biosynthesis, revealed a central role of this pathway in plastid genesis and division. Taken together, these results indicate that peptidoglycans are integral parts of plastids in the whole plant lineage. Elucidating their biosynthesis and further roles in the function of these organelles is yet to be achieved.

## 1. Introduction

According to the endosymbiotic theory of organelles, cyanobacteria are ancestors of chloroplasts in modern plants. This theory, first formulated by Constantin Mereschkowsky in the early days of the 20th century, was reevaluated by Lynn Margolis in 1970 and has been accepted since that time as the way in which the lineage of *Archaeplastida* emerged (for a detailed history of the endosymbiotic theory see [1]), which led to the evolution of algae and land plants. Irrespective of its clarity and broad-based confirmation, several questions remain unanswered.

One of these questions is regarding the integration of cyanobacteria into ancestral eukaryotic cells during primary endosymbiosis. According to widespread belief, a cyanobacterium was engulfed by an α-Proteobacterium via phagocytosis, and the outer plastidic membrane remained as a remnant of this phagosome (for a summary see [2]). However, current knowledge does not support this assumption. In particular, the lipid composition of the outer membrane of the plastidic envelope is similar to that of the outer membrane of gram-negative bacteria [3]. Therefore, it appears that the original double-membrane structure of cyanobacteria was maintained after endosymbiosis. Consequently, the question arises regarding the fate of the original peptidoglycan (PGN) layer between these membranes during the evolution of chloroplasts.

In the last 20 years, there has been growing evidence that the chloroplasts of green algae and early land plants also possess a PGN layer. Even long before PGN was found between the outer membranes of specific plastids (cyanelles) of the Glaucophytes, which is a small ancestral class of the Archaeplastida of single-celled freshwater algae [4,5,6]. In the first experiments with land plants, it was shown that plastid division of the moss *Physcomitrella patens* was affected by β-lactam antibiotics [7], which are known to inhibit penicillin-binding proteins (PBPs) of bacterial cell wall synthesis. Further experiments have revealed an inhibitory effect on plastid division in mosses and ferns, as well as other antibiotics affecting PGN synthesis [8,9,10]. Analyses of the *Physcomitrella* genome revealed that it encodes homologs of almost all the bacterial genes involved in PGN synthesis (Figure 1). Furthermore, it was found that knockout of one of these genes, *PpMurE* (AB194081), leads to defects in chloroplast division in *Physcomitrella* protonemata [11]. Therefore, it was concluded that moss plastids possess a PGN layer in their envelope like gram-negative cells. This was later confirmed by the specific in vivo coupling of an ethenyl derivative of D-Ala-D-Ala, a dipeptide characteristic of the peptide chain of peptidoglycan (Figure 1), to a fluorophore and its microscopic visualization around moss chloroplasts [12].

In contrast to the described results from cryptogamic plants, indications of plastidic PGN layers in angiosperm cells are missing. Therefore, it was concluded that they were no longer able to synthesize PGN [12]. This conclusion is mainly based on the lack of efficacy of β-lactam antibiotics [7] in tomato cell cultures and the unaffected cellular phenotype of *Arabidopsis* D-Ala-D-Ala ligase (*DDL*) mutant chloroplasts. Instead, the loss of this gene in *Physcomitrella* leads to giant chloroplasts [12] because DDL catalyzes the last step of PGN monomer synthesis (Figure 1). Another *Arabidopsis* gene within this pathway, *AtMurE,* is referred to as an additional argument for a lack of PGN synthesis in angiosperms, as the pale-green seedling phenotype of its knockout could not be complemented by its *Physcomitrella* homolog [11]. This was interpreted to be a change in the function of this protein during evolution. From these and other studies, it was concluded that the cyanobacterial PGN layer was maintained in the chloroplasts of early land plants after endosymbiosis but was lost during the evolution of seed plants [13].

However, recent studies cast doubt on this conclusion. It was shown previously that β-lactam antibiotics can affect the plant growth of different mono- and dicotyledenous species [14,15]. In *Arabidopsis*, these antibiotics have been found to cause changes in root architecture and transcriptome in different stress pathways [16]. According to these studies, β-lactam antibiotics cannot generally be considered ineffective in plants, as concluded before [7]. Another argument for the loss of PGN synthesis ability in higher plants is that in the *Physcomitrella* genome, a full set of PGN synthesis genes is present, which is supposed to be the major prerequisite for plant PGN synthesis, as in mosses [11,12]. In the *Arabidopsis* genome, only four homologs of PGN pathway genes (*MurE* (At1g63680), *MraY* (At4g18270), *MurG* (at1g73740) and *DDL* (At1g08840), Figure 1) were found. It is noteworthy, that all sequenced plant genomes to date harbor at least these four homologs (from here on coined as “4-PGN set”). However, recent studies have revealed that an almost full set of PGN synthesis (9 out of 10 genes except PBP; Figure 1) is encoded in several genomes of mono- and dicotyledenous angiosperms [17,18] (from here on coined “full PGN set-PBP”). Therefore, it is very likely that at least in these angiosperm species with a full PGN set-PBP, PGN is synthesized.

Accordingly, the question remains as to whether higher plant plastids are indeed generally missing a surrounding PGN layer. To address this question, we applied for the first time the click chemistry technique for subcellular visualization of PGN in the chloroplasts of higher plants. Therefore, we focused on chloroplasts of *Nicotiana benthamiana* (a “full PGN set-PBP” species according to our analyses, unpublished results) and *Arabidopsis thaliana* (a “4-PGN set” species). In addition to the click chemistry technique, we applied further histological, genetic, and biochemical approaches to address the question of whether angiosperms are principally able to synthesize PGN and deposit it in their plastidic envelope as previously shown in mosses.

## 2. Materials and Methods

### 2.1. Plant Material and Growth Conditions

Seeds of *Arabidopsis* Col-0, as well as T-DNA insertion lines, analyzed in this study were provided by the Nottingham *Arabidopsis* Stock Centre (University of Nottingham, UK) or the *Arabidopsis* Biological Resource Center (University of Ohio, Columbus, OH, USA). Seedling germination experiments were conducted on solid growth media consisting of ½ MS basal salts with 1% sucrose and 1% phytoagar (Duchefa Biochemie, Haarlem, The Netherlands), including further conditional additions (e.g., D-Cycloserine, amino acid azides). These plates were grown for up to 10 days in growth chambers under a long daylight regime (16 h light, 8 h dark) at 18–20 °C. Adult plants were grown for four weeks on soil in a greenhouse with long daylight regime (16 h light, 8 h dark) at 20–22 °C. For experiments with *N. benthamiana,* the leaves of adult plants were also grown on soil for four weeks in a greenhouse. Protonemal cultures of *P. patens* were kindly provided by Andreas Hiltbrunner (University of Freiburg, Germany) and were cultivated as described before [12].

### 2.2. Genetic Characterization of Arabidopsis Lines

PCR genotyping and RT-PCR analyses of *Arabidopsis* T-DNA lines were performed as described before [19]. Primers used for these analyses and their respective sequences are listed in Appendix A.

### 2.3. Cloning of Plasmid Constructs for Expression of GFP Tagged Proteins in Plants

To obtain coding sequences of the proteins to be expressed in plants, they were either amplified by PCR as described before [19] (for primers used for the amplification of *At-DDL*_ΔN-Term_ and *AmiC_AMIN_* see Appendix A) or they were synthesized by commercial providers (Eurofins, Ebersberg, Germany; Invitrogen, Darmstadt, Germany). All sequences were then PCR-amplified to provide suitable ends for cloning into pENTR/D-TOPO (Invitrogen, Darmstadt, Germany) according to the manufacturer’s protocol. After sequence verification of these entry plasmids, they were recombined into a vector for plant expression with a C-terminal GFP fusion, pH7FWG2 [20]. Therefore, an LR kit (Invitrogen, Darmstadt, Germany) was used according to the manufacturer’s protocol.

### 2.4. In Planta Click Chemistry of Amino Acid Azides with Ethynylated Fluorophores

Azide-modified amino acids and AF488-Alkyne were purchased from Jena Bioscience (Jena, Germany). Atto-514-Alkyne was purchased from ATTO-TEC (Siegen, Germany) and the Click-iT cell reaction buffer kit from Invitrogen (Darmstadt, Germany).

To visualize PGN in *Physcomitrella patens,* protonema cells were incubated in BCD medium with ADA (0.25 mM) for 22 h. The cells were washed once with BCD medium and incubated in BCD medium mixed with a Click-iT cell reaction cocktail (1:1) containing Atto-514 (1–5 µM) for 2 h. Cells were washed once with the medium prior to imaging with a confocal microscope.

For click chemistry reactions of *A. thaliana* and *N. benthamiana,* adult plants leaves were infiltrated with 1/2 MS medium containing 0.25 mM azide-modified D-amino acids or 0.125 mM azide modified L-amino acids and incubated for 24–48 h. Infiltrated leaves were either cut off (*A. thaliana*) or partially cut out (*N. benthamiana*) and incubated in ½ MS medium mixed with Click-iT cell reaction cocktail (1:1) containing Atto-514 or AF488 (1–5 µM) for 1.5–2 h. The leaves were washed once with the medium prior to imaging. In the experiments with *A. thaliana* seedlings, all incubation and washing steps were performed as a whole.

### 2.5. Confocal Microscopy

The Fluorophores were imaged using a Leica TCS SP8 AOBS FLIM (Leica Microsystems, Wetzlar, Germany) or Zeiss LSM880 Airyscan (Carl Zeiss Microscopy, Oberkochen, Germany) with a 63X/NA1.2 water objective. Fluorophores were excited (ex) and emission (em) was detected in sequential line scanning mode (with HyD detectors using Leica TCS SP8): GFP and AF-488 (ex/em, 488 nm/490–535 nm), Chlorophyll B (ex/em, 488 nm/690–760 nm), Atto-514 (ex/em, 514 nm/520–560 nm), and RFP (ex/em, 561 nm/590–645 nm). The pinholes were adjusted to 1 Airy unit for each wavelength. Image processing has been described elsewhere [21].

### 2.6. Transient Transformation of Plasmid Constructs in N. benthamiana and A. thaliana

The infiltration of tobacco leaves with-plant expression vectors via *Agrobacterium* mediated gene transfer has been described elsewhere before [21]. *Agrobacterium*-mediated transformation of *A. thaliana* seedlings was based on their cocultivation according to the AGROBEST method [22]. For visualization of chloroplasts with PGN-detecting proteins, *atmurE* mutant seedlings were additionally cocultivated with Agrobacteria harboring the plastid marker CD3-999 pt-rk [23]. For transformation, *Agrobacterium tumefaciens* strains GV3101 and AGL-1 were used.

### 2.7. Biochemical Detection of PGN

For recombinant expression of AmiC2-cat in *E. coli* BL21, the plasmid pET28a-AmiC2cat was kindly provided by Karl Forchhammer (University of Tübingen, Germany), and the expression procedure was performed as described elsewhere [24]. *Arabidopsis* chloroplasts were isolated as described before [25]. PGN from *E. coli* and *A. thaliana* were isolated according to the protocol described by Bertsche and Gust [26]. The PGN binding assay was performed as described before [27,28].

## 3. Results

### 3.1. Visualization of PGN Elements around the Chloroplast of Higher Plants by Click Chemistry

In the initial experiments on the effects of antibiotics on *Arabidopsis* seed germination, D-cycloserine was found to be an effective inhibitor at micromolar concentrations (Appendix A). D-cycloserine is known to be an antibiotic which affects bacterial PGN synthesis by inhibiting D-alanine:D-alanine ligase (DDL) [29]. This led to the question of whether the germination effects of this compound in *Arabidopsis* can be explained by the inhibition of PGN layer formation around chloroplasts, which has been previously demonstrated in mosses [12]. In this study evidence was provided that knockout lines of the *Ddl* gene in the moss *Physcomitrella patens* were no longer able to dimerize D-alanine for peptidoglycan biosynthesis (Figure 1), which led to the formation of macrochloroplasts due to defective plastid division. In contrast, chloroplasts in the cells of *Arabidopsis DDL* KO lines *ddl-1* (SALK 092419C) and *ddl-2* (SAIL 906_E06) remained unaffected, which was explained by the lack of PGN synthesis in higher plants [30]. As this would have been in conflict with the interpretation of our D-cycloserine experiments in *Arabidopsis*, we analyzed the DDL transcripts in the *At-DDL* KO lines *ddl-1* and *ddl-2* in more detail. Remarkably, both lines, which harbor a T-DNA insertion in the first half of the gene, produced transcripts of the 3′-end of *At-DDL* (Appendix A). *At-DDL*, like its homolog from *Physcomitrella* [12], encodes for a fusion gene consisting of two putative DDL enzymes. Interestingly, sequencing and expression of these remaining transcripts from *ddl-1* and *ddl-2* revealed a translatable coding sequence including a start codon, which would encode for a putative full DDL enzyme (*At-DDL*_ΔN-Term_; Appendix A). The expression of a C-terminal GFP fusion of *At-DDL*_ΔN-Term_ in *N. benthamiana* led to the localization of the protein in the cytosol (Appendix A). These findings suggest that the lack of impaired plastid division observed for the previously analyzed *ddl-1* and *ddl-2* mutants might result from the remaining activity of the truncated and cytosolic protein. It is remarkable in this respect that the fusion of two DDL genes is found throughout the whole plant lineage (Appendix A).

To find further evidence for our hypothesis, we decided to visualize different canonical amino acids of bacterial PGN in *Arabidopsis thaliana* and *Nicotiana benthamiana* cells. For this purpose, we incubated the plants with azido-amino acids and a reaction mixture containing ethynylated fluorescent dyes. As exemplarily schematized for the reaction of azido-D-Ala (ADA) and ATTO 514-alkyne in Figure 2A, the reaction mix contains Cu^2+^ ions catalyzing in vivo the coupling of the amino acid azide to the ethynylated fluorescent dye in a copper-assisted cycloaddition, also known as “click chemistry” reaction. This technique has been previously established in bacteria to allow visualization of their cell wall [31]. In a proof-of-principle experiment, moss protonema were incubated with azido-D-Ala (ADA) and ATTO 514-alkyne, as the PGN layer around *Physcomitrella* chloroplasts was detected with a similar approach using ethynyl-D-Ala-D-Ala and Alexa 488-azide in a click chemistry reaction [12]. As shown in Appendix A, our approach also led to characteristic ring-like structures around moss chloroplasts under a confocal fluorescence microscope. 

When *Arabidopsis* Col-0 seedlings were incubated with an adopted click chemistry protocol using ADA and ATTO 514-alkyne, fluorescent ring-like structures were also detected around the chloroplasts (Figure 2B, Appendix A). This indicates that higher plants are generally able to synthesize PGN and possess chloroplasts surrounded by a PGN layer, irrespective of whether they possess a “full PGN set-PBP” of synthesis genes of bacterial ancestry or just a “4-PGN set”. In this regard, it is remarkable that some of the fluorescent signals seem to cross the chloroplasts, like a division plane. Incubation of seedlings with the reaction mix alone with ADA or ATTO 514-alkyne did not result in the characteristic subcellular distribution of fluorescence (Appendix A). To support our findings, two additional azides of canonical PGN amino acids were used instead of ADA in a click chemistry reaction with *Arabidopsis* seedlings, azido-L-alanine (ALA) and azido-L-lysine (ALL). In both cases, a fluorescent ring appeared around the chloroplasts under a fluorescence microscope (Figure 2C,D). To exclude the possibility of non-specific binding or accumulation around the chloroplasts, ADA was substituted in click reactions with azido-D-lysine (ADL), an amino acid derivative usually not found in bacterial PGN. In this case, the fluorescent signal was diffuse and did not accumulate around the chloroplasts (Figure 2E). This image was reminiscent of the treatment of the seedlings solely with the fluorescent dye (Appendix A), which supports the hypothesis that the fluorescent ring-like structures in Figure 2B–D indeed represents PGN around the chloroplast in *Arabidopsis* cells. In click chemistry reaction experiments with D-cycloserine-treated Col-0 seedlings, fluorescence accumulation around the chloroplasts almost vanished (Appendix A, Appendix A). This supports the assumption that PGN synthesis is inhibited by this antibiotic in plants, which leads to the loss of the plastidic PGN layer.

To find out whether PGN biosynthesis and localization around chloroplasts is a common feature in angiosperm plants, we repeated the click chemistry reactions described above in adult leaves of *Nicotiana benthamiana*, a species with a genome encoding a “full PGN set-PBP”. Fluorescence microscopy revealed a similar ring-like structure around the chloroplasts, as observed previously in *Arabidopsis* seedlings. ADA, ALA, and ALL led to fluorescent signals around the chloroplasts (Figure 2F–H), whereas this phenomenon was not be observed when ADL was applied (Figure 2I). This supports our hypothesis that chloroplasts of angiosperms are commonly surrounded by a PGN layer irrespective of their PGN synthesis genes in their genomes.

### 3.2. PGN-Detecting Proteins Accumulate around Chloroplasts and Detect Chloroplastic PGN Biochemically

To visualize plant PGN by alternative means and to investigate how similar plant and bacterial PGNs are, we examined the expression of two PGN recognition proteins (PGRPs) in plants. PGRPs form a superfamily of defense proteins found in the animal kingdom, from insects to mammals, which belong to the innate immune system against pathogenic bacteria [32]. For this purpose, we selected *DmPGRP-SA* from *Drosophila melanogaster* (NM_132499.3) and human *HsPGLYRP-1* (NM_005091.3). For intracellular expression in plants, both sequences were synthesized with optimized *Arabidopsis* codon usage and cloned into appropriate vectors for in planta expression with a C-terminal GFP tag but without N-terminal secretory signal peptides. Both constructs, *DmPGRP_ΔSP_-GFP* and *HsPGRP_ΔSP_-GFP* (sequences are given in Appendix A), missing the encoding sequences for the first 26 and 21 amino acids, respectively, were transiently transformed into *N. benthamiana* leaves. Confocal fluorescence microscopy revealed the accumulation of both PGRP-GFP fusion proteins around the chloroplasts in *N. benthamiana* (Figure 3A,B). The transient expression of these constructs in *Arabidopsis* Col-0 seedlings led to similar results (Figure 3C,D). The almost exclusive localization of both proteins in and around the chloroplasts of wild-type plants indicated a strong affinity of both PGRPs for structures in the plastidic envelope. 

To support this finding, we examined the expression of another type of PGN-detecting protein. Therefore, we chose the N-terminal AMIN domain of the N-acetylmuramyl-L-alanine amidase AmiC from *E. coli*. Its homolog from *Nostoc* has previously been shown to interact specifically with PGN [27]. GFP fusion proteins of the AMIN domain of AmiC from *E. coli* revealed the same accumulation around *Arabidopsis* and *N. benthamiana* chloroplasts after transient expression of *AmiC_AMIN_-GFP* (Figure 3E,F), as previously shown for PGRP-GFP fusion proteins (Figure 3A–D). 

In the amino acid sequences of all applied PGRPs and AmiC, no apparent chloroplast transit peptide was detected in silico. To exclude plastidic import during transient expression of PGN-detecting proteins, we applied a PGN binding assay. Such an assay was used before to show the affinity of the AMIN domain of AmiC from *E. coli* to PGN [27]. To perform a PGN binding assay on chloroplastic PGN, we chose the catalytic domain (AmiC2cat) of AmiC2 from *Nostoc punctiforme* as the PGN-binding protein. As it can be seen in Figure 4, this protein precipitated together with bacterial PGN from *E. coli*. The same occurred when PGN from isolated *Arabidopsis* chloroplasts was added. No precipitation was detected either without the addition of PGN or when instead of AmiC2, a non-PGN-detecting protein, AtDAT1 from *Arabidopsis* [19], was added to the PGN fractions.

Altogether, these results support the observation that PGN also exists in the chloroplast envelope of higher plants. Furthermore, the recognition of chloroplastic PGN by the PGN-detecting proteins used in this study indicates that PGN from plants might be relatively similar to original bacterial PGN.

### 3.3. Knock-Out of AtMurE, a Gene in the PGN Biosynthesis Pathway, Leads to Loss of PGN Layer

To corroborate the hypothesis of PGN biosynthesis in higher plants, we analyzed *Arabidopsis* mutants defective in the putative PGN biosynthesis pathway genes. *At-DDL* is one of these candidates, but knockout lines of this gene have been described as phenotypically unaffected [12]. A possible reason for this lack of phenotype could be incomplete inactivation of the gene (Appendix A), as described above. This assumption was supported insofar as fluorescence microscopy studies of *atddl-1* mutant seedlings after the click reaction with ADA and ATTO 514-alkyne revealed fluorescent rings around their chloroplasts (Appendix A), similar to Col-0 seedlings (Figure 2B). Whether PGN synthesis remains intact in this mutant due to incomplete inactivation, or whether there are other, yet uncharacterized, enzymes responsible for it, is a question that remains to be answered.

In addition to *At-*DDL, we focused on *At*MurE as another candidate enzyme in the plant PGN biosynthesis pathway. Mutants of this gene were described as showing a pale-green phenotype with impaired thylakoid development. Additionally, a knock-out of *murE* in *Physcomitrella* led to formation of giant chloroplasts [33]. To evaluate the function of AtMurE in PGN synthesis in higher plants, we analyzed several knockout mutants of this gene. Therefore, we chose two previously characterized *atmurE* null alleles, *atmurE-3* and *atmurE-4* [33] and isolated and analyzed an uncharacterized mutant allele, *atmurE-5* (pde316, CS16226) [34]. According to the flanking sequences of the borders of the latter T-DNA insertion, mutant parts of the first exon and the whole second exon of *atmurE* were deleted (Figure 5A). Loss of function of the *murE* gene could be confirmed in this line insofar as the pale-green seedling phenotype of homozygous *murE-5* plants was similar to that of other *murE* mutants [33], and no *murE* transcript was detected by RT-PCR in this line (Figure 5B,C).

In the cotyledon cells of all three analyzed *atmurE* null alleles, clear chloroplast structures were no longer visible by brightfield microscopy (Figure 5D, Appendix A). The impression of loss of plastidic structure was supported by click chemistry reactions with seedlings of *atmurE-5*. None of the amino acid azides applied before clear structures around chloroplasts could be visualized in the cells (Figure 5E–G) as in Col-0 wild-type cells before (Figure 2B–D). Additionally, when *DmPGRP_ΔSP_-GFP* and *HsPGRP_ΔSP_-GFP* were transiently expressed in *atmurE-3* seedlings, their characteristic accumulation around chloroplasts disappeared. In addition, no chloroplast structures were recognized with the co-expressed plastid marker (Appendix A). Therefore, we concluded that loss of AtMurE function interrupts PGN synthesis in these mutants, leading to the loss of clear distinguishable plastidic structures.

## 4. Discussion

In the present study, several independent lines of evidence showed that angiosperms are also able to synthesize PGN to surround their chloroplasts. In click chemistry reactions with azide-derivatized D-amino acids and ethenylated fluorophores, PGN-like structures around the chloroplasts of *Arabidopsis thaliana* and *Nicotiana benthamiana* could be visualized. The use of canonical D- and L-amino acid derivatives for bacterial PGN in these experiments indicated conserved peptide structures of these plastidic PGN isoforms. This conclusion was supported by the intracellular expression of PGN-detecting proteins that accumulated in and around chloroplasts, and subsequent PGN binding assays. Treatment with the PGN synthesis inhibitor D-cycloserine, as well as loss of the PGN synthesis gene *AtMurE,* led to severe growth impairment of *Arabidopsis* seedlings with destructured chloroplasts. The macroscopic and microscopic phenotypes in both cases indicated an essential function of chloroplastic PGN in chloroplast biogenesis and plant development. According to the presented cytological, genetic, and biochemical data, it can be postulated that the former bacterial PGN layer was maintained after endosymbiosis in plastidic outer envelopes throughout whole plant evolution. This suggests that a PGN layer in the plastidic envelope is likely to be found in all plants.

Therefore, several observations, results, and hypotheses regarding PGN biosynthesis genes and chloroplast division in plants must be reassessed. One of these findings is the inefficacy of β-lactam antibiotics in higher plants, as this is one of the major arguments against their ability to synthesize PGN. Although studies on the effects of β-lactam antibiotics on higher plants are rather rare, it was shown that their rate of accumulation in carrot and lettuce is relatively low [35,36]. It is still not clear whether this effect is due to low uptake, biodegradation, or both. However, this serves as a possible explanation for the small effects of β-lactam antibiotics on higher plants. Another possibility that must be considered is the lack of homologs of penicillin-binding proteins (PBPs) in seed plant genomes. Although in all flowering plant genomes with “full PGN set-PBP” a coding region for a transglycosylase domain like in PBP was found, in none of these genomes a transpeptidase domain protein, like the bacterial ones, could be identified [18] to bind to β-lactam antibiotics.

Another finding to be reassessed is the functional assignment of DDL proteins in angiosperms. Because of the normal chloroplast division in the cells of *Arabidopsis DDL-1* KO lines in contrast to the giant chloroplasts observed in *Physcomitrella DDL* KO lines, it was postulated that this protein was originally a D-Ala-D-Ala ligase and changed its function during plant evolution [12]. Our results revealed incomplete knockout in the *Arabidopsis* T-DNA lines. The remaining transcripts encode DDL-like gene products, which would explain this difference. Although our observations of *Arabidopsis* seedlings treated with the DDL inhibitor D-Cycloserine supported this interpretation, experimental confirmation is still needed, particularly because this DDL-like gene product localizes to the cytosol instead of plastids. It seems that this mislocalization At-DDL_ΔN-Term_ has no apparent impact on PGN synthesis and deposition to the plastidic envelope. To resolve this contradiction, it has yet to be shown whether this partial DDL-like gene product, or even the entire AtDDL protein, are still functional D-Ala-D-Ala ligases.

In addition, the function of AtMurE must be interpreted newly in the light of the results of our study. Initially, the defective chloroplast phenotype of *Physcomitrella murE* mutants could not be rescued by its homologs from the gymnosperm *Larix gmelinii* and the angiosperm *Arabidopsis thaliana* [33,37]. This was considered as evidence for a shift of MurE function towards a plastid-encoded RNA polymerase in the evolution of plants after the split of lycophytes [13,38]. Our findings regarding MurE function are in conflict with these studies, but this may be explained by different substrate specificities. In a recent report [39], it was shown that MurE from *Anabaena* and *Physcomitrella* specifically incorporated meso-Diaminopimelate (DAP) into muropeptides instead of L-Lysine, which we detected in higher plant PGN in our experiments (Figure 3D,H). It is noteworthy that cyanobacteria as plastidic ancestors usually integrate DAP at this position [40], and that plant MurE, like other bacterial PGN synthesis homologs in plants, does not seem to be of cyanobacterial origin [41,42]. Additionally, a second, yet uncharacterized, MurE isoform in all cryptogamic plants with higher homology to gymnosperm and angiosperm homologs has been reported [39]. Future experiments will show whether both MurE isoforms also differ in their amino acid specificity, and whether this is the reason for the failed complementation of *Physcomitrella* mutants by higher plant MurE.

Although our study confirmed the presence of a PGN layer in the chloroplast envelopes in angiosperm plants, several questions remain to be answered. For example, the microscopic and molecular structures of plastidic PGN layers must be investigated in detail. In mosses, this layer is rather thin and close to the inner envelope membrane. First, it was not possible to detect it in the chloroplast membranes of *Physcomitrella* by electron microscopy [43], but later by single-pixel densitometry of such electron micrographs [44]. The molecular structure of angiosperm plant PGN was revealed to be similar to that of its bacterial isoform, because it was detected in our experiments with azides of canonical amino acids of bacterial PGN and with proteins detecting bacterial PGN. This conclusion needs to be confirmed in the future by the extraction, isolation, and structural resolution of plant PGN, which is a long way to go, as the recent approaches in *Physcomitrella* showed [39].

Nevertheless, detection of PGN in *Nicotiana benthamiana* and *Arabidopsis thaliana*, possessing a “full PGN set-PBP” and “4-PGN set”, respectively, implies the general ability of angiosperms to produce PGN, irrespective of their genetic setup. Furthermore, this conflicts with the hypothesis that loss of PGN synthesis in higher plants may be one of the prerequisites for the development of polyplastidity and plastidic diversity in seed plants [45,46]. However, simultaneously, it poses the question of how a plant with just four out of the initially ten PGN synthesis genes of bacterial origin is able to maintain the plastidic PGN layer. With regard to our results, we postulate that in angiosperms, a second pathway for PGN synthesis evolved with the genes of the “4-PGN set” as an essential core. Hypothetically, species with the “full PGN set-PBP” would possess the original synthesis pathway, whereas in plants with the “4-PGN set” the novel alternative pathway should be present. Analyzing mutants of *MraY*, *MurG* and *DDL* would be helpful to verify this theory. Interestingly, within one plant order, representatives with both pathway setups were found, for example in Brassicales with *A. thaliana* (“4-PGN set”) and *Carica papaya* (“full PGN set-PBP”) or in Rosales with *Fragaria vesca* (“4-PGN set”) and *Malus domestica* (“full PGN set-PBP”) [17,18]. The appearance of both PGN pathway types within closely related lineages implies convergent loss of genes in the “4-PGN set” species and supports the hypothesis of an alternative PGN synthesis pathway. Next, PGN in chloroplasts of other angiosperm species should be identified and compared with respect to their genetic PGN pathway inventory to verify this theory. This will also serve as a starting point for the identification of genes and proteins involved in a putative novel PGN biosynthesis pathway.

Another major issue regarding the plastidic PGN layer is its role in plant growth and development. In cyanobacteria, as the accepted ancestor of plastids in Viridiplantae [42], multiple functions have been assigned to its PGN layer (for a summary see [40]), which may serve as starting points for further investigations in this regard. It has been shown before [47] that cyanobacterial PGN plays a major role in cell division. Our observations of D-cycloserine treated wild-type *Arabidopsis* seedlings and of untreated *atmurE* mutants revealed structural defects in the chloroplasts of these plants. Nevertheless, these defective chloroplasts cannot be clearly defined as giant chloroplasts. Such giant chloroplasts were previously shown in algae [48,49], mosses [10], and ferns [8,9] treated with cell wall antibiotics. Furthermore, they were observed in moss mutants in the PGN synthesis pathway [12,50] and *Arabidopsis* mutants of FtsZ and MinE, which are two proteins of bacterial origin involved in plastid division [51,52]. It has been demonstrated before that FtsZ proteins form a network around moss plastids coined as “plastoskeleton” [53,54,55] in analogy to the cytoskeleton. Because of the conserved association of bacterial PGN with FtsZ [56], it is tempting to speculate that the plastidic PGN layer is essential for plastid division in all plants. Therefore, one focus of future research of PGN functionality in angiosperms should be on division of chloroplasts. Alternatively, structural stabilization of chloroplasts should be considered a major property of the plastidic PGN layers. It is interesting in this regard that MurG and MraY were found to interact in *E. coli* [57]. Furthermore, MurG and MurE were shown to interact in *Thermotoga maritima* [58] and *Bordetella pertussis* [59]. As homologs of all three proteins are found in the complete plant lineage, their interaction behavior will be also matter of future research. Especially, the interaction of MurG and MraY was shown to be in the center of the bacterial divisome and elongasome (for a summary see [60]), implying a similar role for both proteins in chloroplast biogenesis. Recently, it has been suggested [61] that the PGN layer, in combination with the plastoskeleton, contributes to the integrity and mechanical stability of chloroplasts. The reason for this chloroplast stabilization is the protection of plastid swelling and damage due to the osmotic imbalance between the cytosol and organelles. The maintenance of chloroplast integrity is therefore a prerequisite for its optimal function, and therefore essential for plant life, which would be a general property in all plants according to our findings.

## Figures and Tables

**Figure 1 cells-12-00563-f001:**
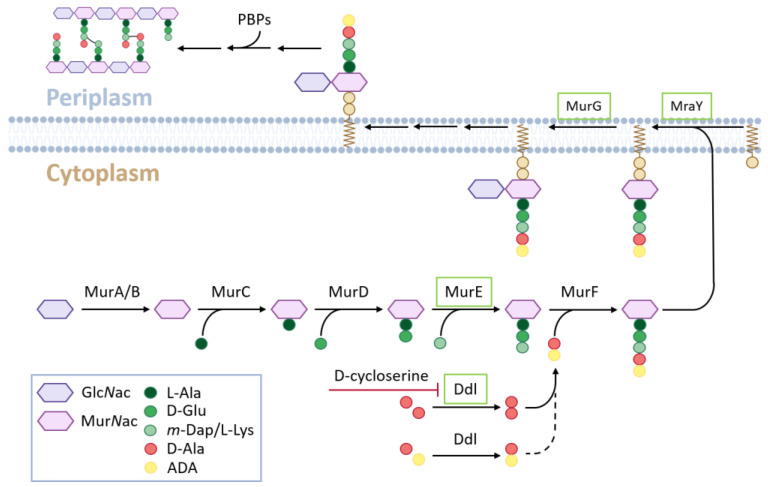
Principal scheme of PGN biosynthesis. Enzymes of the “4-PGN set” are boxed in green.

**Figure 2 cells-12-00563-f002:**
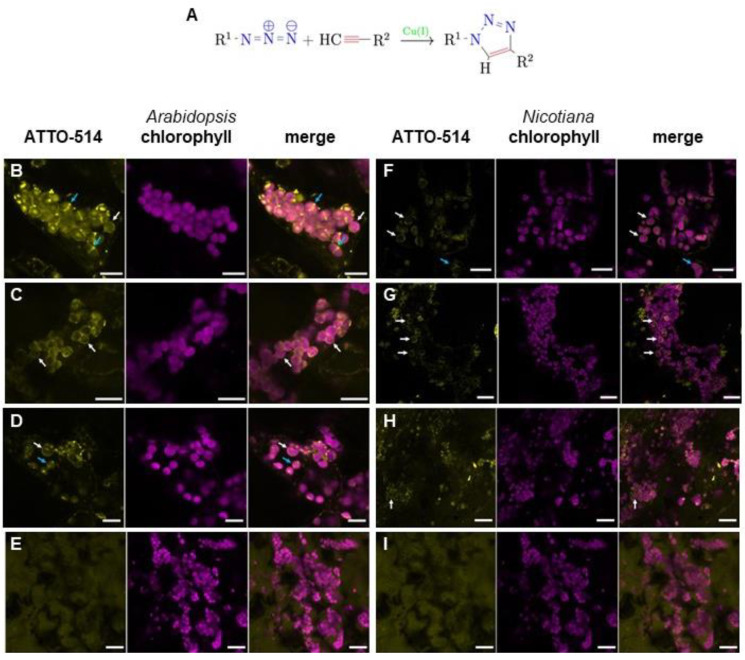
Chloroplasts of *Arabidopsis thaliana* and *Nicotiana benthamiana* cells are surrounded by PGN-like structures. (**A**) General coupling scheme of azide amino acid derivatives (R^1^) with ethenylated fluorescent dyes (R^2^) via click chemistry as used in this study. (**B**–**E**) Fluorescence microscopic images of *Arabidopsis* cells after click chemistry reactions of ATTO 514-Alkyne with (**B**) ADA (scale bar 10 µm), (**C**) ALA (scale bar 15 µm), (**D**) ALL (scale bar 10 µm), and (**E**) ADL (scale bar 20 µm). (**F**–**I**) Fluorescence microscopic images of tobacco cells after click chemistry reactions of ATTO 514-Alkyne with (**F**) ADA (scale bar 15 µm), (**G**) ALA (scale bar 20 µm), (**H**) ALL (scale bar 20 µm), and (**I**) ADL (scale bar 20 µm). Left picture in a series of three pictures is always ATTO-514 fluorescence (ATTO-514), middle picture is chlorophyll autofluorescence (chlorophyll), and right picture is the merged image. White arrows indicate chloroplasts surrounded by fluorescent ring-like structures, blue arrows highlight structures crossing chloroplasts like a division plane.

**Figure 3 cells-12-00563-f003:**
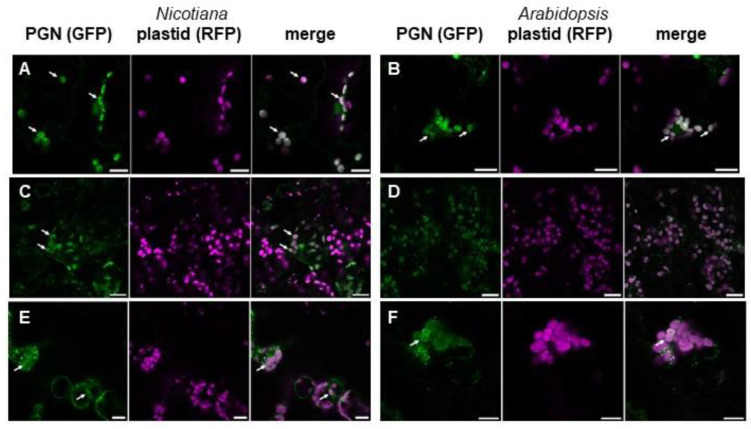
PGN-recognizing proteins localize in and around chloroplasts of *Nicotiana* and *Arabidopsis* cells. a-b, Fluorescence microscopic image of *Nicotiana* cells expressing (**A**) *DmPGRP_ΔSP_-GFP* and (**B**) *HsPGRP_ΔSP_-GFP* (green) together with a plastid marker (magenta; scale bars 15 µm). (**C**,**D**) Fluorescence microscopic image of *Arabidopsis* cells expressing (**C**) *DmPGRP_ΔSP_-GFP* and (**D**) *HsPGRP_ΔSP_-GFP* (green) together with a plastid marker (magenta; scale bars 20 µm). (**E**,**F**) Fluorescence microscopic image of (**E**) *Nicotiana* (scale bar 15 µm) and (**F**) *Arabidopsis* cells (scale bar 10 µm) transiently expressing *AmiC_AMIN_-GFP* (green) together with a plastid marker (magenta). The first picture in a series of three pictures is always the PGN detecting protein fused to GFP (PGN (GFP)), the second one the plastid marker fused to RFP (plastid (RFP)), and the third one represents the merged image. White arrows indicate chloroplasts surrounded by fluorescent ring-like structures.

**Figure 4 cells-12-00563-f004:**
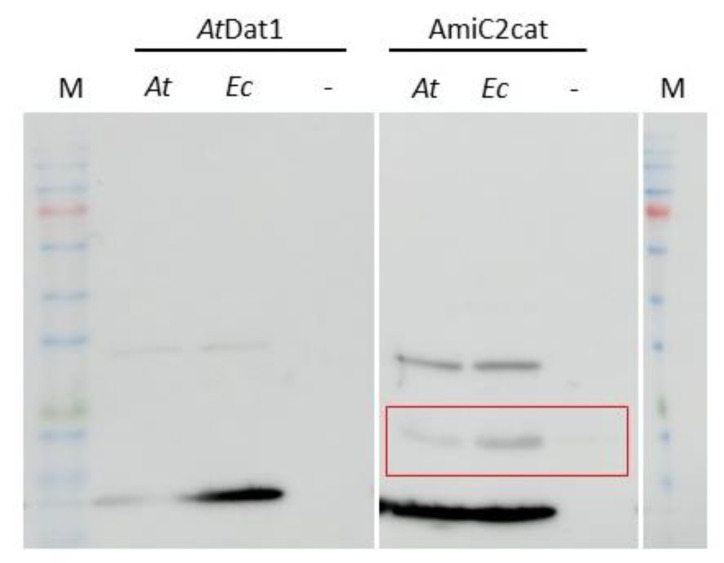
PGN from *E. coli* and *Arabidopsis* chloroplasts lead to precipitation of AmiC2cat in PGN binding assays. In the left panel bound AtDAT1, and in the right panel bound AmiC2cat in PGN preparations from *A. thaliana* (*At*), *E. coli* (*Ec*) and with no added PGN (–) are shown. Precipitated AminC2cat by PGN is marked by a red box.

**Figure 5 cells-12-00563-f005:**
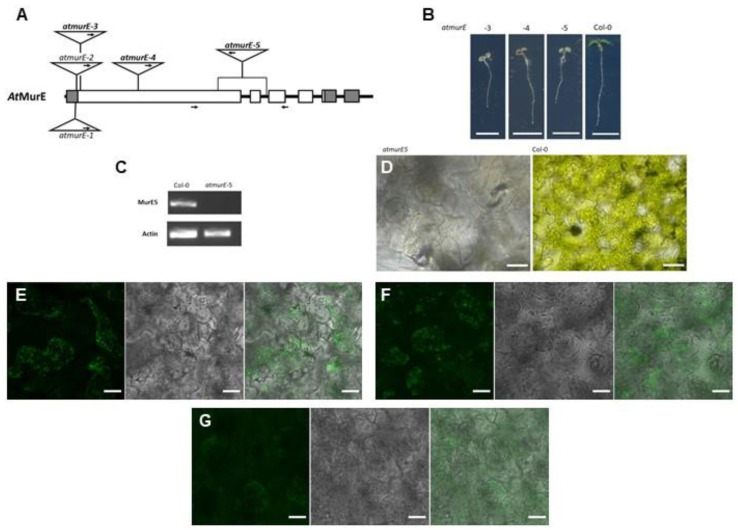
AtMurE is involved in PGN synthesis. (**A**) Schematic representation of the *AtMurE* gene and the position of the T-DNA insertion sites in *atmurE-1* to *atmurE-5.* Exons are represented by boxes and introns by connecting thick lines. Sequences encoding 5′- and 3′-untranslated regions are marked in grey. The insertions of *atmurE-3, atmurE-4* and *atmurE-5*, used in this study, are highlighted in bold. Arrows in the triangles indicate position of left border in the insertions. Arrows under gene scheme mark the positions of primers used for RT-PCR. (**B**) Phenotype of *atmurE-3, atmurE-4, atmurE-5,* and corresponding wild-type seedlings. Scale bar 5 mm. (**C**) RT-PCR analysis of *AtMurE* in *atmurE-5* and corresponding wild-type seedlings. (**D**) Bright field microscopic images of cells from *atmurE-5* and corresponding wild-type cotyledons (scale bar 30 µm). (**E**–**G**) Fluorescence microscopic images of *atmurE-5* cells after click chemistry reactions of ATTO 514-Alkyne with (**E**) ADA, (**F**) ALA, and (**G**) ALL. Left picture in a series of three pictures is always ATTO-514 fluorescence (ATTO-514), middle picture is chlorophyll autofluorescence (chlorophyll), and right picture is the merged image (scale bars 20 µm).

## Data Availability

Not applicable.

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
