# Peer review of "The Chloroplast Envelope of Angiosperms Contains a Peptidoglycan Layer"

_cells, 2023, doi:10.3390/cells12040563_

Round 1

Reviewer 1 Report

The manuscript adds evidence for the presence of peptidoglycan-like components on the outer surface of chloroplasts of angiosperms. These components should be evolutionarily related to the peptidoglycans of the endosymbiont cyanobacteria, that evolved to present day chloroplasts, but their presence in angiosperms is intriguing.

Microscopies of different leaf preparations of Arabidopsis, Nicotiana are compared with previous publications reporting peptidoglycans surrounding chloroplasts of mosses. Assays with T-DNA modified Arabidopsis and GFP labelling suggest that chloroplasts of angiosperms do contain peptidoglycan-like components. The evolutionary origin, and the structure and functions of these components in angiosperms are cautiously discussed by authors that recognize that the preliminary results require further detailed investigations. Some remarks should be advanced to the present manuscript.

Authors should refer the pioneering publications, as early as the nineties, dealing on the peptidoglycan of the chloroplast/cyanelle of the glaucophyte Cyanophora paradoxa.

Figures 5D-G and S6A-B are hardly informative. Not at all about chloroplasts. Authors may consider suppression. What amorphous chloroplasts are (line 337)? If maintained, insert size scale in S6A-B.

The information provided by microscopy is limited, authors should consider analyses with purified isolated chloroplasts.

Among the hypothesis postulated for peptidoglycan-like of angiosperms, the manuscript should point potential interactions with the extensively investigated components of the chloroplast division machinery.

Author Response

Dear Reviewer,

first of all, I would like to thank you for the valuable and constructive comments on the manuscript. They will be of great value to improve it. You will find our responses to your suggestions and comments in red in a point for point fashion.

The manuscript adds evidence for the presence of peptidoglycan-like components on the outer surface of chloroplasts of angiosperms. These components should be evolutionarily related to the peptidoglycans of the endosymbiont cyanobacteria, that evolved to present day chloroplasts, but their presence in angiosperms is intriguing.

Microscopies of different leaf preparations of Arabidopsis, Nicotiana are compared with previous publications reporting peptidoglycans surrounding chloroplasts of mosses. Assays with T-DNA modified Arabidopsis and GFP labelling suggest that chloroplasts of angiosperms do contain peptidoglycan-like components. The evolutionary origin, and the structure and functions of these components in angiosperms are cautiously discussed by authors that recognize that the preliminary results require further detailed investigations. Some remarks should be advanced to the present manuscript.

Authors should refer the pioneering publications, as early as the nineties, dealing on the peptidoglycan of the chloroplast/cyanelle of the glaucophyte Cyanophora paradoxa.

Early publications on the characterization of peptidoglycan in cyanelles were added to the introduction.

Figures 5D-G and S6A-B are hardly informative. Not at all about chloroplasts. Authors may consider suppression. What amorphous chloroplasts are (line 337)? If maintained, insert size scale in S6A-B.

I agree with the reviewer that these figures just provide evidence of aberrant chloroplasts in murE mutants. But they show that peptidoglycan biosynthesis does not take place in these mutants. This was the aim of these experiments as also the title of Fig. 5 says. The term “amorphous chloroplasts” was deleted and replaced by descriptions of the observations. Size scales were integrated in S6A-B.

The information provided by microscopy is limited, authors should consider analyses with purified isolated chloroplasts.

In respect to this suggestion, we must admit that we encountered different technical problems in working with isolated chloroplasts. In our first attempts to do click chemistry with isolated chloroplasts we noticed that just newly dividing chloroplasts were incorporating the click substrates, and therefore the efficiency was too low to be analyzed. Isolation of in planta clicked chloroplasts also resulted in too small number of analyzable chloroplasts. Instead, we treated chloroplasts with lysozyme and analyzed them cytometrically with FACS. We can detect a change of chloroplast shape, but these are preliminary data, and the works are still in progress.

As indicated in Materials and Methods we even performed the biochemical detection of peptidoglycan with isolated Arabidopsis chloroplasts. Additionally, we also tried the peptidoglycan isolation from chloroplasts, but the isolated amounts were not sufficient for further chemical analysis. Recently, there was a method published for the isolation and analysis of peptidoglycan from mosses (Dowson et al. 2022) which we are currently planning to implement to chloroplasts of higher plants.

Among the hypothesis postulated for peptidoglycan-like of angiosperms, the manuscript should point potential interactions with the extensively investigated components of the chloroplast division machinery.

I added a section within the last paragraph on the interaction of MurG, MurE and MraY in bacteria, their functions in bacterial divisome and elongasome, and their homologs in plants and their putative functions.

Reviewer 2 Report

The plant cell is a unique object in which two two-membrane compartments function, often performing opposite functions. And if the evolution of mitochondria is devoted to many works on animals and humans, then the origin of chloroplasts until recent years was quite speculative and was based on conjectures and theories. However, with the development of modern research methods, reliable data were obtained that the precursors of chloroplasts were not internalized by the plant cell through endocytosis. In the work of the authors, it is very clearly and thoroughly shown that the outer membrane of chloroplasts contains peptidoglycan, which could not possibly exist if this membrane had a plasma membrane origin. It should be noted that similar works have already been published earlier, however, the methods used by the authors are used for these purposes for the first time. This unarguably makes it possible to comprehensively obtain more reliable confirmation of the problem raised in the authors' article - the origin and evolution of chloroplasts. Unfortunately, the authors do not sufficiently discuss in their article the pathways of penetration of chloroplast precursors, which helped not to be digested by the host cell, but to gain a foothold as a permanent and inseparable compartment. I would like the authors to suggest possible ways and factors that contributed to this in the discussion. However, the absence of such a large section is not a significant drawback for an experimental article, and can only be considered as a recommendation to the authors.  I would recommend the authors in the photo in figures 2,3,5 to add captions for each channel and "merge" for the summing photo.

Author Response

Dear Reviewer,

first of all, I would like to thank you for the valuable and constructive comments on the manuscript. They will be of great value to improve it. You will find our responses to your suggestions and comments in red in a point for point fashion.

The plant cell is a unique object in which two two-membrane compartments function, often performing opposite functions. And if the evolution of mitochondria is devoted to many works on animals and humans, then the origin of chloroplasts until recent years was quite speculative and was based on conjectures and theories. However, with the development of modern research methods, reliable data were obtained that the precursors of chloroplasts were not internalized by the plant cell through endocytosis. In the work of the authors, it is very clearly and thoroughly shown that the outer membrane of chloroplasts contains peptidoglycan, which could not possibly exist if this membrane had a plasma membrane origin. It should be noted that similar works have already been published earlier, however, the methods used by the authors are used for these purposes for the first time. This unarguably makes it possible to comprehensively obtain more reliable confirmation of the problem raised in the authors' article - the origin and evolution of chloroplasts.

Unfortunately, the authors do not sufficiently discuss in their article the pathways of penetration of chloroplast precursors, which helped not to be digested by the host cell, but to gain a foothold as a permanent and inseparable compartment. I would like the authors to suggest possible ways and factors that contributed to this in the discussion. However, the absence of such a large section is not a significant drawback for an experimental article and can only be considered as a recommendation to the authors.

Although I agree with the reviewer´s point of view about the importance of the penetration mechanism and the “survival” process during endosymbiosis, I decided not to add an additional section about these subjects to this manuscript. The reason for this decision lies in the focus of this study to show peptidoglycan in the envelope of angiosperms. The existence of peptidoglycan in the envelope of its ancestors has been shown before, where a discussion of endosymbiotic processes would fit far better in my opinion. Furthermore, we extended the section in our revised manuscript about the putative functions of peptidoglycan and its synthesis genes in chloroplast biogenesis. These arguments imply, at least partially, how the integration of these organelles proceeded and why these elements were maintained.

I would recommend the authors in the photo in figures 2,3,5 to add captions for each channel and "merge" for the summing photo.

Descriptions of the different channels have been added to the captions of all three figures. Additionally, we added titles to series of images in Fig. 2 and 3 to recognize quicker the individual channels.